

# Particle Size Dependence of Biogenic Secondary Organic Aerosol Molecular Composition

Peijun Tu[1], Murray V. Johnston[2]

[1,2]Department of Chemistry and Biochemistry, University of Delaware, Newark, Delaware 19716, USA

*Correspondence to*: Murray V. Johnston (mvj@udel.edu )

## Abstract

Formation of secondary organic aerosol (SOA) is initiated by the oxidation of volatile organic compounds (VOCs) in the gas phase. Mass transfer to the particle phase is thought to occur primarily by a combination of condensation of non-volatile products and partitioning of semi-volatile products, though particle phase chemistry may also play a role if it transforms

semi-volatile reactants into non-volatile products. In principle, changes in particle composition as a function of particle size allow the relative contributions of e.g. condensation (a surface-limited process) and particle phase reaction (a volume-limited process) to be distinguished. In this work, SOA was produced by β-pinene ozonolysis in a flow tube reactor. Aerosol exiting the reactor was size-selected with a differential mobility analyser, and individual particle sizes between 35 and 110 nm in diameter were characterized by on- and off- line mass spectrometry. Both the average oxygen-to-carbon (O/C) ratio and

carbon oxidation state (OSc) were found to decrease with increasing particle size, while the relative signal intensity of oligomers increased with increasing particle size. These results are consistent with oligomer formation in the particle phase i.e. accretion reactions, which become more favoured as the surface-to-volume ratio of the particle decreases. Analysis of a series of polydisperse SOA samples showed similar dependencies: as the mass loading increased (and average surface-to-volume ratio decreased), the average O/C ratio and OSc decreased while the relative intensity of oligomer ions increased. The

results illustrate the potential impact that particle phase chemistry can have on biogenic SOA formation and the particle size range where this chemistry becomes important.

## 1 Introduction

Ultrafine particles, defined here as smaller than 100 nm in diameter, constitute the largest number of particles in the atmosphere and are of interest owing to their disproportionate influence on climate and human health (Bzdek et al., 2012;

Zhang et al., 2012). Particularly important is their role in formation of cloud condensation nuclei (CCN) and their corresponding impact on radiative forcing (Myhre et al., 2013). For small ultrafine particles to grow to a climatically-relevant size, the particle growth rate must exceed the loss rate. The greatest uncertainty associated with particle growth and it impact





on radiative forcing is the contribution of secondary organic matter (Carslaw et al., 2013), which is formed by oxidation of volatile compounds in the gas phase followed by subsequent migration of the products to the particle phase.

On a global scale, organic aerosol constitutes a substantial fraction of the total aerosol mass in the atmosphere (Hallquist et al., 2009; Jimenez et al., 2009; Kanakidou et al., 2005) and most of this is secondary (Jimenez et al., 2009; Ng et

al., 2010). On a molecular level, ambient organic aerosol is very complex, encompassing hundreds to thousands of individual compounds (Chan et al., 2013; Goldstein and Galbally, 2007; Hallquist et al., 2009; Mutzel et al., 2015; Putman et al., 2012). Laboratory secondary organic aerosol (SOA) is similarly complex and a variety of oxidation and degradation pathways have been proposed to explain the product distributions (Daumit et al., 2013; Hall and Johnston, 2012; Hallquist et al., 2009; Perraud et al., 2012). Part of this complexity arises from the formation of high molecular weight (MW) oligomeric species from two

or more precursor molecules (Kalberer et al., 2004; Tolocka et al., 2004). Oligomers can constitute almost half of the SOA mass in laboratory experiments (Gao et al., 2004; Hall and Johnston, 2011).

Ultrafine particle growth is thought to proceed mainly by a combination of condensation and partitioning. Condensation occurs when the gas phase mixing ratio of a compound is larger than its equilibrium vapour pressure and the "on" rate determined by collisions of gas phase molecules with the particle surface exceeds the "off" rate determined by re-

evaporation of molecules from the particle phase, whereas partitioning occurs when the gas phase mixing ratio is smaller than the equilibrium vapour pressure and the compound distributes between the gas and particle phases (Pankow, 1994). The recent detection and characterization of extremely low-volatility organic compounds (ELVOCs) in the gas phase has uncovered a previously underappreciated pathway for condensational growth (Bianchi, 2016; Ehn et al., 2014). For monoterpene oxidation, the range of ELVOC species includes both highly functionalized monomers and oligomers.

The dependence of chemical composition on particle size can provide insight into particle growth mechanisms. Processes limited by the amount of available surface area, such as condensation, are favoured in smaller particles where the surface-to-volume ratio is high. Processes limited by the amount of available volume, such as partitioning, are favoured in larger particles where the surface-to-volume ratio is low. Superimposed on these dependencies is the radius-of-curvature (Kelvin) effect on molecular volatility, which also favours the incorporation of lower volatility species into smaller diameter

particles. Winkler et al. (Winkler et al., 2012) have reported size-resolved composition of particles between 10 and 40 nm in diameter that were produced by α-pinene ozonolysis. Based on signal intensities of species detected by thermal desorption chemical ionization mass spectrometry (TDCIMS), 10-20 nm particles contained a greater fraction of lower volatility species, while 30-40 nm particles contained a greater fraction of higher volatility species. In this experiment, species volatility was qualitatively assessed by the loss of signal intensity due to sample evaporation over time. In second study of SOA from α-

pinene ozonolysis, Zhao et al. (Zhao et al., 2013) used measurements of gas phase molecular species by chemical ionization mass spectrometry to show a positive correlation of higher MW (and presumably less volatile) species with the number



concentration of 10-20 nm particles, whereas lower MW (and presumably more volatile) species were positively correlated with the number concentration of 30-40 nm particles. In a third study of SOA from α-pinene ozonolysis, Kidd et al. (Kidd et al., 2014) showed that particles in the 250-500 nm range contained a greater fraction of oligomers while particles greater than 500 nm contained a greater fraction of monomers. Molecular composition measurements by Zhao et al. (Zhao et al., 2015) of size-selected particles produced by trans-3-hexene ozonolysis showed that particles smaller than 100 nm contained a greater fraction of high molecular weight (MW) oligomers than particles larger than 100 nm. All of the above experiments are consistent with the concept that higher MW, lower volatility species formed in the gas phase are more strongly represented in smaller diameter particles, as would be expected a condensation-driven process.

Particle phase chemistry, specifically accretion reactions in the particle phase that form higher MW lower volatility oligomers from higher volatility lower MW monomers (Barsanti and Pankow, 2005), have been proposed as an additional pathway for SOA formation. Accretion chemistry represents an alternative source to ELVOC condensation for oligomers that are detected in the particle phase. It has been noted that relatively few ELVOC molecular formulas obtained from gas phase measurements match those of oligomers detected in particle phase measurements (Mentel et al., 2015; Tu et al., 2016). This dissimilarity could arise from subsequent reaction of ELVOCs after they enter the particle phase, or by the formation of completely new oligomers in the particle phase through accretion chemistry. In principle, these two sources can be distinguished through the size dependence of particle composition, since molecular species derived from condensation (surface area limited) should be more strongly represented in smaller particles while those derived from accretion chemistry (volume limited) should be more strongly represented in larger particles. A particle size dependent molecular composition arising from particle phase reaction was suggested in a modelling study of SOA produced from dodecane photooxidation, though experimental measurements in that work were confined to particle size distributions as a function of reaction time (Shiraiwa et al., 2013).

In the work presented here, we consider the particle size-dependent chemical composition of SOA produced by β-pinene ozonolysis. The particles of interest, 30-110 nm in diameter, are large enough that the radius-of-curvature effect of molecular volatility is negligible and size-dependent changes in composition reflect the relative importance of surface area vs. volume limited processes. In addition, the particles are small enough that phase separation within the particles is unlikely to occur (Veghte et al., 2013a). The results provide direct evidence for accretion chemistry as a significant source of oligomers in SOA from this precursor.



## Experimental Section

### 2.1 Aerosol Generation and Size Selection

Figure 1 shows the experimental setup used in this work. All gas flows were generated from zero air (model 737, Aadco Instruments Inc., Miami, FL, USA) to minimize contamination. SOA was generated in a flow tube reactor (FTR) (section A of Fig. 1) described previously (Hall et al., 2013; Tu et al., 2016). In most experiments, the concentrations of β-pinene and ozone after mixing in the reactor were 1 ppmv and 10 ppmv respectively, giving an SOA mass loading of about 2300 µg/m$^3$ at the reactor exit. In a separate set of experiments, the SOA mass loading was varied in the 5-2300 µg/m$^3$ range by varying the β-pinene concentration between 0.03 and 1 ppmv. Blank samples were obtained by flowing zero air into FTR to mix with ozone in the absence of β-pinene. All FTR experiments were performed at a low relative humidity (8%).

In a control experiment, polydisperse SOA from the FTR (2300 µg/m$^3$) that had been previously collected onto a filter was extracted into 50/50 acetonitrile/water and atomized (ATM 226, TOPAS, Dresden, Germany) to produce a control aerosol. Section B of Fig. 1 shows the apparatus, which included a diffusion dryer to reduce the amount of water vapour in the aerosol flow. The goal of this experiment was to generate SOA-like aerosol that did not have a particle size dependent chemical composition associated with it, though the composition of this aerosol was not expected to be precisely the same as the original collected SOA because of possible chemical reactions prior to and/or during atomization. This experiment also provided the opportunity to assess possible artefacts due to sample collection and analysis after size selection (see section 2.2). The particle size distribution of the control aerosol was fine tuned to be as similar as possible to SOA from the FTR by varying the gas flow conditions into the atomizer and the concentration of extracted SOA in the solution used for atomization (Stabile et al., 2013). Sample blanks for the control experiment were obtained by atomizing pure solvent (Park et al., 2012).

Particle size distributions were monitored with a Scanning Mobility Particle Sizer (SMPS, TSI Incorporated, St. Paul, Minnesota, USA). Specific particle sizes within the size distribution were selected with a separate Differential Mobility Analyzer (DMA, model 3081, TSI Incorporated, St. Paul, Minnesota, USA). The mobility diameters studied in this work were 35, 60, 85, 110 nm. Size distributions are shown in Fig. S1. Mass concentrations are given in Table S1. Because of the low mass concentrations after size selection, zero-air was sent through the entire experimental apparatus for 12 h after each experiment to remove contamination.

### 2.2 Sample Collection and Off-line Analysis with HR-MS

Particles were collected with a Nano Aerosol Sampler operating in the spot collection mode (NAS-s; Section D in Fig. 1). The sampler was custom designed and built by Aerosol Dynamics, Inc. (Berkeley, California, USA). This device uses a water-based condensation method (Hering and Stolzenburg, 2005) to either sample particles into a small spot in a collection well or to concentrate them into an outlet flow aerosol flow. When used in spot mode, particles deposited in the collection



well were able be dissolved in a minimum amount of solvent for subsequent offline analysis. In this device, aerosol first passed through a "conditioner" region at 5℃, followed by a heated "initiator" region at 35℃ where the aerosol became saturated with water vapour. The aerosol then entered a cooled "moderator" region at 10℃ where the temperature decrease created a supersaturated vapour. In this region, water condensed on the particles to produce droplets, which were subsequently

focused to a ~1 mm spot in a collection well. The well was heated to 35℃ to evaporate the condensed water (dry deposition mode) from collected particles. After a sufficient amount of sample was collected (~10 μg in these experiments), acetonitrile/deionized water (1:1) solvent was added to the well to dissolve the collected particles to a concentration of 0.1 mg/mL for offline analysis. Depending on aerosol mass concentration, 2-48 hours were required to collect a sufficient amount of sample (see Table S1). Mass spectra of polydisperse SOA were similar for samples collected with NAS-c at 35℃ vs. a

standard filter at room temperature, confirming that gentle heating inside the NAS-c did not cause thermal decomposition of this aerosol. NAS-c had the advantage over filter collection of being able to work with smaller sample sizes and therefore shorter collection times.

      Offline molecular characterization was performed by high resolution mass spectrometry (HR-MS) with a Q Exactive™ Hybrid Quadrupole-Orbitrap Mass Spectrometer (Thermo Scientific, Waltham, Massachusetts, USA) coupled to

a heated-electrospray ionization (HESI) probe. For these measurements, the sample flow rate was 3 μL/min with an injection volume of 10 μL. Other operating parameters included: spray voltage, 2.5-3.5 kV; capillary temperature, 250-275˚C. The possibility of producing artefacts of non-covalently bound clusters was ruled out using the approach discussed elsewhere (Hall et al., 2013; Tu et al., 2016). Full MS scans were acquired over the range 100−1000 m/z with a mass resolving power of 70 000. Each spectrum was obtained by averaging ~130 scans over a period of approximately 1.0 min and then processed with

Xcalibur™ Software.

      Five replicates were analyzed for each sample, and molecular formulas had to be positively detected and assigned in all five replicates in order to be considered further. Blank sample subtraction was performed for each sample prior to the data analysis. Data analysis was performed as described previously targeting closed-shell molecular formulas (Tu et al., 2016). We also performed Kendrick Mass Defect plots and RDB-O (Herzsprung et al., 2014; Phungsai et al., 2016) value (the

maximum RDB-O distribution resides in -10 and 10 for hydrophobic CHO species (Herzsprung et al., 2014)) as the updated criterions to help filtering the unreasonable assigned formulas. After removing background peaks, unreasonable formulas and redundancies due to isotopic substitution, hundreds of unique molecular formulas remained for each sample. The number of unique formulas and mass and intensity weighted average O/C ratios of these formulas are given for each experiment in Table S2.



## 2.3 On-line Analysis with NAMS

Online single particle analysis was performed with a modified Nano Aerosol Mass Spectrometer (NAMS) shown in section C of Fig. 1. The previously reported NAMS configuration for analysis of 10-30 nm diameter particles (Ross Pennington and Johnston, 2012; Wang and Johnston, 2006; Wang et al., 2006) was modified to enable analysis of particles between 40
and 110 nm in diameter. Particles entered the mass spectrometer through an aerodynamic lens assembly that focused particles into a tight beam in the ion source region. A focused, high energy pulsed laser beam (532 nm, 5 Hz, 230 mJ/pulse focused to an effective spot size of about 0.1 mm dia.) intercepted the particle beam. When a particle was in the beam path when the laser fired, a plasma was formed that quantitatively disintegrated the particle into multiply charged atomic ions, whose relative signal intensities gave the elemental composition of the particle (Klems and Johnston, 2013; Zordan et al., 2010). Aerosol
mass spectra were obtained by averaging ~200 individual particle spectra, and the process of obtaining an average mass spectrum was repeated three times over the course of each experiment, which provided confirmation that particle composition did not change during an experiment. Figure S2 gives an example mass spectrum of size selected 60 nm SOA particles. Table S3 summarizes the elemental composition data from the various experiments.

## 3 Results and Discussion

### 3.1 Elemental Composition of Size-Selected SOA

NAMS measurements provided the opportunity to determine the bulk elemental composition of size-selected SOA, specifically the oxygen-to-carbon (O/C) ratio. O/C ratios as a function of particle size are summarized in Table S3 and shown in Fig. 2a. The trend of decreasing O/C ratio with increasing particle size is consistent with the expectation that lower volatility (and more highly functionalized) molecules are preferentially found in small particles where the high surface-to-volume ratio
favours condensation over partitioning, while higher volatility (and less functionalized) molecules are preferentially found in large particles where the lower surface-to-volume ratio favours partitioning to a greater degree. This same general trend was found for molecular analysis, as shown in Fig. 2a by the mass and intensity weighted O/C ratios (Hall et al., 2013) averaged over all assigned molecular formulas. For both positive and negative molecular ions, the average O/C ratio was also found to decrease with increasing particle size. The average O/C ratios obtained from negative ions were much greater than those from
positive ions, which reflects the bias of negative ion detection toward molecules containing acid groups whereas positive ion detection is biased toward detection of molecules containing carbonyls only (Hall et al., 2013). The O/C ratios obtained from NAMS lie between the positive and negative mode O/C ratios obtained from HR-MS, which is reasonable since the NAMS measurements represent all molecular species in the sample.



The similarity of the NAMS and HR-MS data in Fig. 2a suggests that these size dependencies are not artefacts of the respective measurement methods. To provide further confirmation, the control aerosol was analysed by the same procedure. The results are summarized in Table S3 and shown in Fig. 2b. As expected, no particle size dependence is observed in either the NAMS or HR-MS data. Furthermore, the (particle size independent) O/C ratio for negative ion mode of the control aerosol in Fig. 2b matches the average O/C ratio of the polydisperse sample (shown as a line in Fig. 2a). This similarity is expected since the negative ion mode preferentially detects highly oxidized molecules that have low volatility and are likely to be retained through the extraction and atomization steps used to generate the control aerosol. The O/C ratio for positive ion mode of the control aerosol in Fig. 2b is somewhat higher than the average O/C ratio for polydisperse aerosol in Fig. 2a. This difference suggests that lower O/C ratio (and presumably higher volatility) molecules are lost during the extraction and atomization process. Loss of these molecules is consistent with the NAMS O/C ratio for the control aerosol in Fig. 2b, which is higher than the NAMS O/C ratio for polydisperse aerosol in Fig. 2a and matches the O/C ratio for the negative ion mode.

### 3.2 Molecular Composition of Size-Selected SOA

To better understand particle size-dependent changes in chemical composition on a molecular level, it is helpful to consider the products of β-pinene ozonolysis in the gas phase and their relevance to chemical processes that drive particle growth. Aerosol yields for β-pinene ozonolysis under conditions similar to the experiments performed here have been reported to be on the order of 30%, with products spanning a wide range of volatilities (von Hessberg et al., 2009). First and foremost with regard to particle formation in an unseeded experiment is the production of ELVOCs, which for monoterpene ozonolysis (Ehn et al., 2014) have molecular formulas spanning monomers (for the purpose of this study defined as molecules having fewer than 9 carbon atoms since β-pinene ozonolysis results in the loss of a carbon atom) and dimers (defined here as molecules having between 10 and 18 carbon atoms). Monomer ELVOCs necessarily are highly oxidized since this is needed to achieve very low volatility, whereas dimers can be somewhat less oxidized owing to their larger molecular size. ELVOCs corresponding to higher order oligomers (defined here as molecules having greater than 18 carbon atoms) are exceedingly rare, which is not surprising since the probability of gas phase reaction decreases quickly with increasing number of precursor molecules. ELVOCs are inefficiently produced from β-pinene ozonolysis owing to its exocyclic double bond, with an estimated yield less than 0.1% (Ehn et al., 2014). Most products of β-pinene ozonolysis are volatile or semi-volatile monomers partition between the gas and particle phases (Jaoui and Kamens, 2003).

Molecular level changes in particle composition are summarized in Fig. 3 and Fig. 4, which compare the mass spectra and corresponding molecular products, respectively, for both positive and negative ion mass analysis of size-selected SOA at 35 nm and 110 nm. These changes are interpreted on the basis of monomers (which encompass both ELVOCs that condense and semi-volatile products that partition), dimers (which encompass both ELVOCs that condense and products of accretion





chemistry that are produced directly in the particle phase), and higher order oligomers (produced almost exclusively by accretion chemistry in the particle phase).

Figure 4 shows plots of average carbon oxidation state (OSc) vs. number of carbon atoms for all assigned molecular formulas, colour coded to indicate formulas that are unique to 35 nm particles, unique to 110 particles, and common to both
particle sizes. OSc for a molecular formula is defined as (Kroll et al., 2011):

$$OSc = 2 \, O/C - H/C$$

where O/C and H/C are the oxygen-to-carbon and hydrogen-to-carbon ratios of the formula. Two general trends are observed in Fig. 4 for both ion polarities. First, the unique formulas in 35 nm particles tend to have higher OSc than the common formulas, while the unique formulas in 110 nm particles tend to have lower OSc than the common formulas. Second, 110 nm
particles tend to have a greater number of unique oligomer formulas with greater than 18 carbon atoms than 35 nm particles, and this disparity increases with increasing number of carbon atoms. Both of these differences are consistent with the elemental composition changes in Fig. 2. Higher OSc formulas tend to have higher O/C elemental ratios, which favour higher O/C in smaller particles. Formulas of higher order oligomers tend to have lower OSc than monomers and dimers, which favour lower O/C in larger particles.

Further insight can be gained from the ion signal intensities of each assigned molecular formula. Figure 5 shows the fraction of the total signal intensity due to higher order oligomers (defined here as formulas with greater than 18 carbon atoms) as a function of particle size. These oligomers are produced almost exclusively by accretion chemistry in the particle phase (Barsanti and Pankow, 2005, 2006). In Fig. 5, both ion polarities show an approximate linear increase of oligomer intensity with increasing particle diameter. A linear increase is expected for a volume-limited process such as accretion chemistry
relative to a surface-limited process such as condensation. Here, "volume-limited" means that the particle volume available for reaction increases linearly with increasing particle volume i.e. the cube of the particle diameter. It does not preclude the possibility that processes such as phase separation or hindered diffusion within the particle causes a portion of the total particle volume to be inaccessible to this chemistry, though we note that phase separation is unlikely in such small particles (Veghte et al., 2013a). Oligomerization has been considered for many years to be a significant contributor to SOA formation (Hall and
Johnston, 2011; Kalberer et al., 2004; Kroll and Seinfeld, 2008; Tolocka et al., 2004; Trump and Donahue, 2014), and Fig. 5 shows through chemical measurement that this contribution strongly depends on particle size.

The particle size dependence of oligomers reported here for β-pinene SOA is opposite that reported previously in the ultrafine size range for two other precursors, α-cedrene and trans-3-hexene (Zhao et al., 2013, 2015). This difference is related most likely to the origin of the species involved (gas phase vs. particle phase). The molecular structure of α-cedrene is much
more conducive to the production of ELVOCs in the gas phase than β-pinene, making condensational growth more likely. In contrast, the analysis of β-pinene SOA in Fig. 5 focuses on accretion chemistry and does not include "dimer" ($C_{10}$ to $C_{17}$)



products of β-pinene ozonolysis, which may contribute to growth by a combination of condensation from the gas phase and accretion chemistry in the particle phase. The dimer products of β-pinene ozonolysis in this work show a roughly constant relative signal intensity with increasing particle size, which likely reflects the multiple sources of these species. High molecular weight oligomers observed in the trans-3-hexene ozonolysis experiment were suggested to be formed in the gas phase by

reaction of a peroxy radical with the stable Criegee intermediate, which is unlikely for either α-cedrene or β-pinene (Zhao et al., 2015).

Figure 6 shows the intensity-weighted, average OSc of all monomer formulas (carbon number < 10) as a function of particle size. The average OSc for species detected in negative ion mode is essentially independent of particle diameter, suggesting very little change in composition among these species. This dependence is suggested by the OSc (-) vs. carbon

number plot in Fig. 4, where most monomer species are found to be common to both particle sizes. The lack of a composition dependence is not surprising since negative ion mode is biased toward detection of more highly oxidized species whose rates of condensation relative to each other are not expected to be size dependent.

More interesting is the plot in Fig. 6 for positive ion mode, which shows a substantial decrease of monomer OSc with increasing particle diameter. This dependence is suggested by the OSc (+) vs. carbon number plot in Fig. 4 for positive ion

mode, where fewer molecular formulas are common between the two particle sizes and the unique formulas in 35 nm particles have higher OSc than the unique formulas in 110 nm particles. While it is tempting to interpret these data as enhanced partitioning of higher volatility species into larger particles, this explanation is problematic. The particle sizes investigated in this work are too large for the Kelvin effect to influence molecular volatility. To the extent that partitioning reaches equilibrium, there should be no difference in the particle phase concentrations of partitioned species since the equilibrium state

does not depend explicitly on particle size. Partitioning does depend on the relative volumes of the gas and particle phases for the entire system, but these volumes are fixed in the current experiments since the selection of different particle sizes was performed at the same time point in the SOA formation process.

Particle phase reactions such as accretion chemistry have the ability to increase SOA mass by transforming semi-volatile monomers into non-volatile oligomers. Molecular partitioning from the gas phase to the particle phase provides a

continuous source of reactant molecules to feed the reaction as it proceeds. The decreasing OSc(+) of monomers with increasing particle size suggests that oligomerization is partially reversible. Larger particles have greater oligomer content relative to the total particle mass (Fig. 5), so they also have greater potential to yield decomposition products. Oligomer decomposition could be an artefact of the sample preparation and analysis steps after particle collection, or it could be an intrinsic aspect of accretion chemistry that occurs prior to particle collection and analysis. If reversibility is an intrinsic aspect

of accretion chemistry, then the higher amounts of low OSc monomers in large particles suggests that diffusion within the particle phase is hindered (Faulhaber et al., 2009; Grayson et al., 2015; Koop et al., 2011; Renbaum-Wolff et al., 2013) and/or





phase separation has occurred (Laskina et al., 2015; Veghte et al., 2013b; Virtanen et al., 2011; Werner et al., 2016), effectively trapping the released monomers within the particle making them unable to re-equilibrate with the gas phase. Reversibility of the oligomerization process provides a reasonable explanation why β-pinene SOA yields are so strongly dependent on temperature and relative humidity (von Hessberg et al., 2009).

## 3.3 Elemental and Molecular Composition of Polydisperse SOA with Varying Mass Loading

Additional experiments were performed to study the change in composition of polydisperse SOA as a function of mass loading and compare the results to those discussed above for composition as a function of particle size for a single mass loading. Three mass loadings were investigated: 5 µg/m$^3$, 240 µg/m$^3$ and 2300 µg/m$^3$ (which was also the mass loading used to study the particle size dependencies above). The results are included in Tables S1-S3. Mass spectra and plots of OSc vs. carbon number are given in Fig. S3 and Fig. S4. The mass loading trends mirror those of particle size. For both NAMS and HR-MS, the average O/C ratio decreases with increasing mass loading. Unique molecular formulas in the low mass loading spectra generally have high OSc values, while unique molecular formulas in the high mass loading spectra generally have low OSc values. Oligomer ions increase in relative signal intensity with increasing mass loading. These similarities are not surprising since the particle size distribution shifts to larger sizes as the mass loading increases (mode diameter increases from 23 nm at 5 µg/m$^3$ to 76 nm at 2300 µg/m$^3$). Because of the particle size dependence of accretion reactions, high mass loadings also have a higher percentage of oligomer products.

A decrease in average O/C ratio with increasing SOA mass loading has been reported previously for elemental analysis of laboratory SOA produced from related biogenic precursors (Chhabra et al., 2010; Shilling et al., 2009). The results presented here utilizing both elemental and molecular composition measurements provide a mechanistic explanation for this general observation: accretion chemistry increases in importance relative to condensation and partitioning as the particle size increases.

## 4 Conclusions

In this work, elemental and molecular analysis of size-selected biogenic SOA particles between 35 nm and 110 nm diameter is reported. These results provide clear evidence for oligomer formation via accretion chemistry in the particle phase and show that the impact of accretion chemistry (a particle volume-limited process) on molecular composition increases with increasing particle size. Since accretion reactions provide the opportunity to transform semi-volatile monomers into non-volatile oligomers, they represent a chemical pathway to increase the aerosol yield and also potentially to increase the growth rate of ultrafine particles.



Since the ELVOC yield from β-pinene ozonolysis is much smaller than that from other biogenic SOA precursors, it is possible that accretion chemistry plays a larger role in formation of β-pinene SOA than in formation of SOA from other precursors where the likelihood of ELVOC condensation is greater. As noted in a previous study, oligomer formation in β-pinene SOA is strongly dependent on reaction conditions (von Hessberg et al., 2009). The impact of accretion chemistry for may be greater for the laboratory SOA studied here than ambient SOA owing to different reaction conditions. In this regard, elemental analysis shows that ambient SOA is generally more highly oxidized than laboratory SOA (Aiken et al., 2008). Also, molecular analysis of ambient samples show that oligomer ions tend to have very low signal intensities (Mentel et al., 2015), though higher oligomer signal intensities appear to be strongly correlated with CCN activity (Kourtchev et al., 2016). The results we present here illustrate the potential impact that particle phase chemistry can have on SOA formation and the particle size range where this chemistry becomes important.

## Acknowledgements

This research was supported by the U.S. National Science Foundation under grant number CHE-1408455. The authors thank Dr. Andrew Horan for assistance in HR-MS analysis.

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





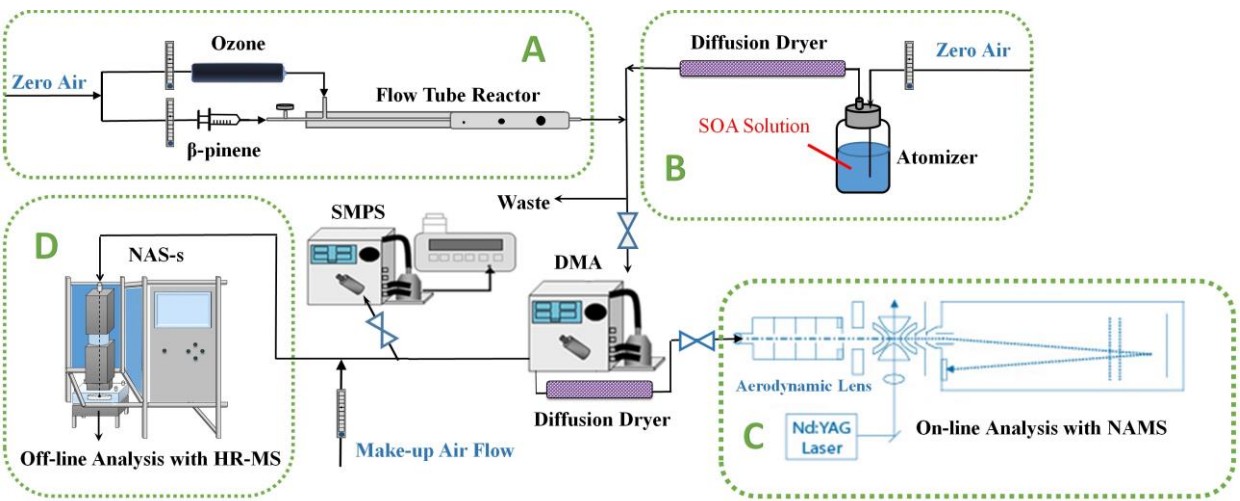

**Figure 1.** Schematic of the experimental workflow. SOA is produced either directly from the flow tube reactor (A) or re-aerosolization from an atomizer (B). Analysis is performed on-line by NAMS (C) or off-line by HR-MS after sample collection with NAS-s (D).





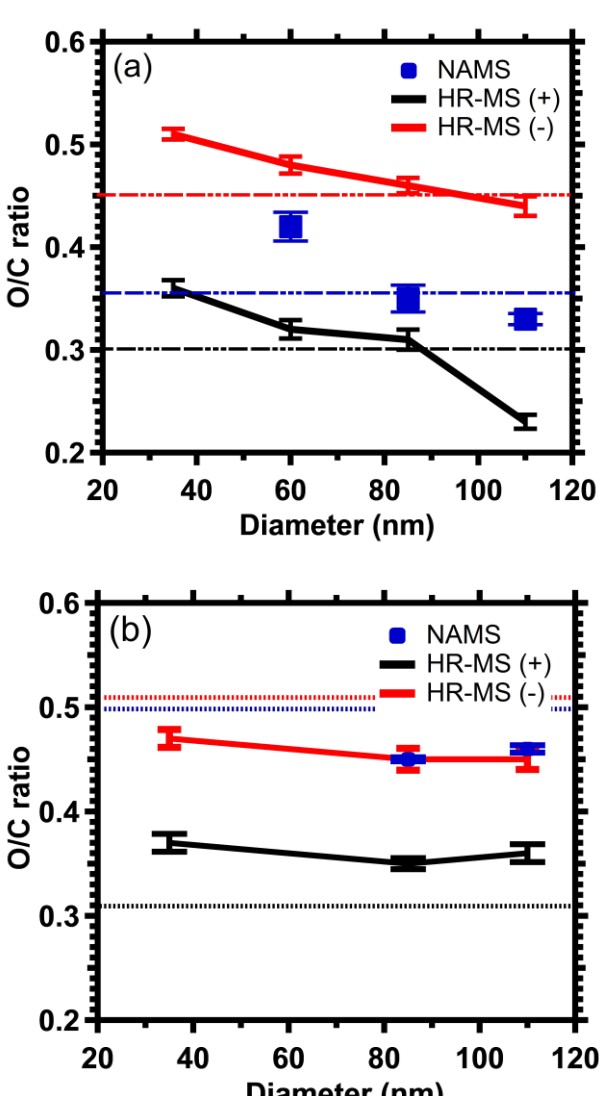

5   **Figure 2.** Average O/C ratio vs. particle diameter for SOA (a) generated from the flow tube reactor and (b) re-aerosolized from the atomizer. Dashed lines in the plots give the O/C ratios of the corresponding polydisperse SOA samples. Error bars represent one standard deviation for the 5 replicate experiments.





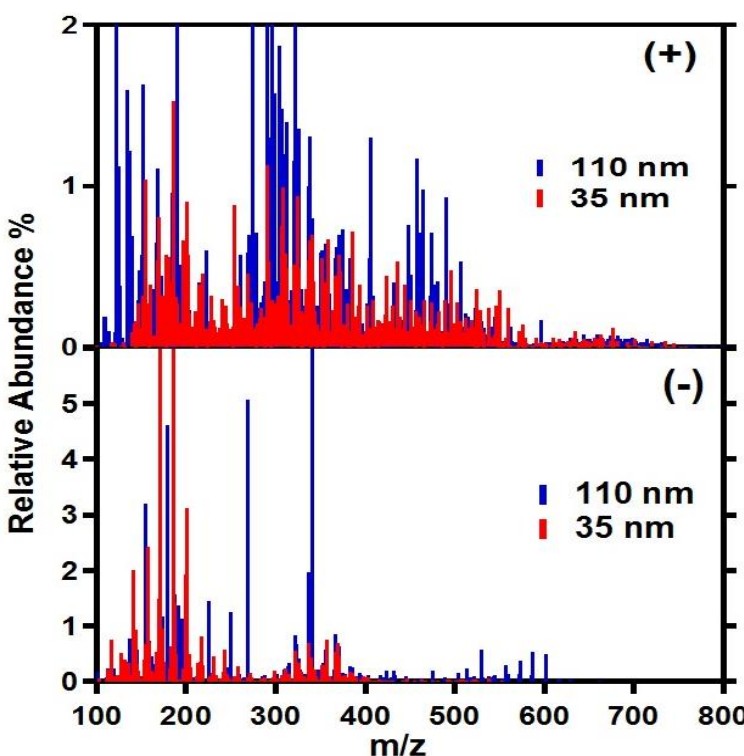

**Figure 3.** Positive (+) and negative (-) ion mass spectra of 35 nm (red) and 110 nm (blue) monodisperse SOA samples averaged over the 5 replicate experiments.





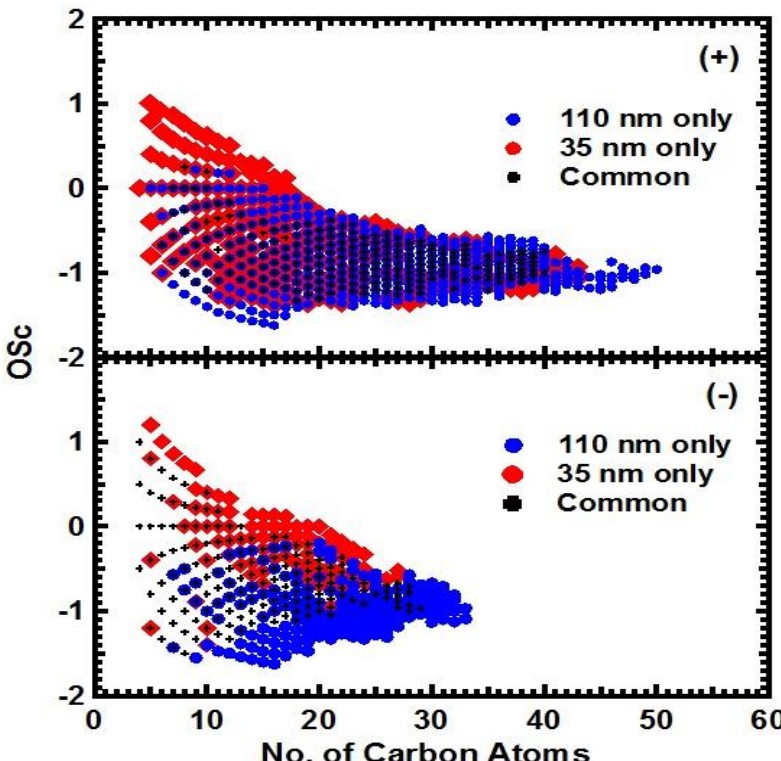

**Figure 4.** Carbon oxidation state (OSc) vs. number of carbon atoms for assigned molecular formulas from the positive (+) and negative (-) ion mass spectra of 35 and 110 nm monodisperse SOA samples. Unique formulas in the 35 nm samples are shown in red. Unique formulas in the 110 nm samples are shown in blue. Formulas common to both samples are shown in black.





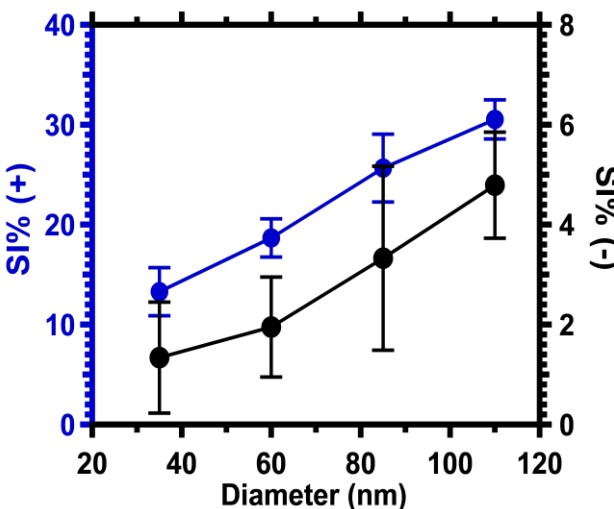

**Figure 5.** Percentage of total signal intensity (SI %) in positive (+) and negative (-) ion mass spectra from higher order oligomers (molecular formulas having greater than 18 carbon atoms) vs. particle diameter. Error bars represent one standard deviation for the 5 replicate experiments.

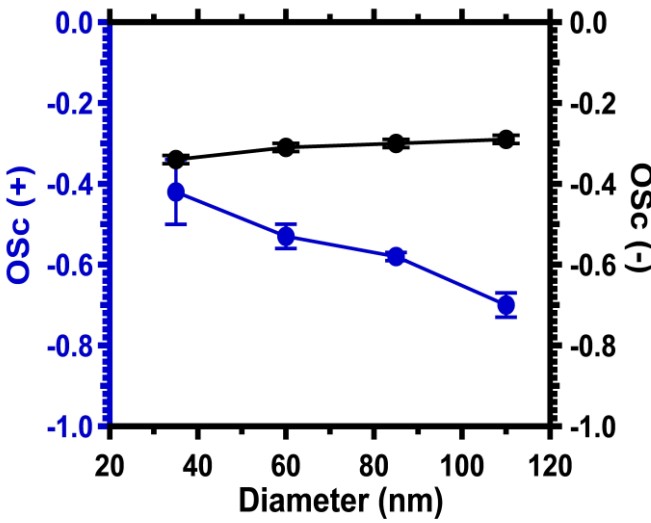

**Figure 6.** Average OSc vs. particle diameter for monomer species (less than 10 carbon atoms in the assigned formula) detected in positive (+) and negative (-) ion mass spectra. Error bars represent one standard deviation for 5 replicate experiments.