# Peer review of "Supporting Information for: Particle Size Dependence of Biogenic Secondary Organic Aerosol Molecular Composition"

_Atmospheric Chemistry and Physics, 2017_

## Referee Comment (RC1) · Anonymous Referee #1 · 25 Feb 2017

General comments: The authors report on the use of nano-aerosol mass spectrometry (NAMS) to elucidate the oxidative extent and oligomeric content of SOA. By measuring these two parameters as a function of particle size, they were able to elucidate the relative contributions of condensation and particle phase reaction to particle growth. Their main conclusion is that accretion reactions become more important for larger particle sizes, as one might expect as the surface-to-volume ratio of the particle decreases. Using NAMS data for a number of particle sizes and chemical systems, the authors present a very convincing case that particle phase chemistry can have a substantial impact on the lifecycle (i.e., formation and aging) of biogenic SOA.

Specific comments: Table S1: The authors show the average mode diameter of the

aerosol to be 76 nm for "Polydisperse a" and 240 nm for "Polydisperse d." These are not "close" in my opinion. Figure S1 seems to support my comment. Distributions from atomization are dramatically different from those of the FTR. Actually, I wonder why this is important. Perhaps a bit more discussion would be helpful. Table S3: What do the errors in the "Unique Molecular Formulas" column of Table S3 indicate if molecular formulas had to be detected and assigned in all five replicates in order to be considered? Page 7, Lines 4-11: This entire paragraph is very confusing. The authors should add some guidance to the reader to ensure that the reader is evaluating the correct data (i.e., state specifically what symbol, or line, is used to represent each of the data sets. So, perhaps write "The O/C ratio for positive ion mode of the control aerosol (black solid line) in Fig. 2b is somewhat higher than the average O/C ratio for polydisperse aerosol (black dashed line) in Fig. 2a" I'm not even sure I got those representations correct. Page 9, Lines 10-13: The authors state that "the lack of composition dependence is not surprising…" I don't understand why this would not be surprising. It may be obvious to the authors, but some clarity and, if possible literature references, should be provided to substantiate their statement. Figure S2: Caption should include particle diameter, assuming these particles have been size selected for 60 nm (as stated in manuscript, Page 12, Line 6) Figure S3: Please provide absolute mass loading for high and low case in caption.

Small typo errors: Page 2, Line 27: "…associated with particle growth and its impact…" Page 3, Line 8: "…as would be expected from a condensation-driven process." Page 5, Line 8: "Depending on aerosol mass concentrations, 2-93 hours were required…" according to Table S1). Page 5, Line 26: "…criteria to help filter the unreasonable…"

---

## Author Comment (AC1) · 26 Mar 2017

We thank the reviewer for excellent comments and suggestions to clarify and improve the manuscript. Reviewer comments and our responses are given below. Line numbers refer to the original manuscript.

Specific comments:

The referee questions the "similarity" of the size distributions for polydisperse samples a and d (Table S1 and Figure S1).

Author response: We recognize the confusion caused by our wording on p. 4 lines 17-19. What we meant to say, and will modify the text accordingly, is that fine tuning
of the size distribution of the control aerosol was done in order to assure that we had a sufficient aerosol mass concentration at each of the mobility sizes of interest (35, 60, 85, 110 nm) to permit chemical analysis. Fine tuning was required because of the difficulty of generating particles at the smallest mobility diameter using our particular atomizer.

The referee asks about the definitions of entries in Table S3.

Author response: We will modify Table S2 to include the following footnote for the "Unique Molecular Formulas" column saying that these values represent the average and standard deviation from five replicate samples. Each replicate gives slightly different results, which is why it is important to perform several replicates. The difference among replicates arises mostly from formulas that have very low signal intensities. We will also add a new column entitled "Common Molecular Formulas" that gives the number of common formulas observed in all five replicates for a given sample type – these are the formulas that are evaluated and discussed in the results and discussion. This will give the reader a better idea of repeatability of the experiment, and will show that most of the assigned formulas are indeed detected in all five replicates.

Referee comment concerning page 7 lines 4-11 and Figures 2a and 2b.

Author response: We will modify the legend and caption of the figures as well as the text to more clearly indicate the identities of the markers (NAMS, HRMS(+) and HRMS(-)), for both size-selected and polydisperse aerosols. (By the way, the referee got the representations correct in their comment.)

Referee comment about page 9, lines 10-12, why is it "not surprising" that there is no composition dependence for negative ions in Fig. 6b?

Author response: We will expand this discussion in the revised manuscript to explain more thoroughly, while at the same time removing the "not surprising" phrase. Our group and others (e.g. Hall 2013, Tu 2016, Mutzel 2015) have noted in the

past that molecular formulas obtained from negative ion spectra are generally more highly oxygenated/oxidized than those obtained from positive ion spectra. Highly oxygenated/oxidized formulas are suggestive of molecules that have very low volatilities. If essentially all of the monomers detected in negative ion spectra are nonvolatile, there will be no particle size dependence in their relative ability to be incorporated into particles – all of these molecules will condense with similar probability when striking the particle surface. In contrast, the next paragraph (p. 9 lines 13-22) discusses positive ion monomers, which potentially have a very wide range of volatiles – some are nonvolatile, while others are semivolatile. The relative amounts of non- vs. semi- volatile monomers will change as a function of particle size for the reasons given in this paragraph.

Referee comment about Figure s2.

Author response: The caption to Figure S2 will be modified – this spectrum is for 60 nm size-selected particles.

Referee comment about Figure S3. Author response: Labels displaying the specific mass loading values will be added.

Typographical changes noted by referee: Author response: All will be corrected as mentioned by the reviewer. Thank you very much for pointing these out.

---

## Referee Comment (RC2) · Anonymous Referee #2 · 7 Apr 2017

"Particle size dependence of biogenic secondary aerosol molecular composition" by Tu and Johnston describes a study in which particles generated from ozone-initiated oxidation of $\beta$-pinene are chemically analyzed using offline and online analysis for both size-resolved and polydispersed samples. Understanding size-resolved chemical composition of nanometer-sized particles has important implications for quantifying and, ultimately, modeling new particle growth following nucleation. The current study mostly confirms prior studies that show that condensation of low volatility organics are important for smaller particles, whereas oligomer formation is important for larger particles. Even though it is not clear that a lot of new ground was broken in this study, it appears to be nicely done and the writing is clear and concise. I have only a few

comments that I wish for the authors to address prior to publication, indexed below according to page/line numbers (I will include minor editorial suggestions along with slightly more substantive issues):

1/26: "small ultrafine" – I suggest removing "small"

2/12: I find the distinctions that the authors make between condensation and partitioning somewhat confusing. It has always been my practice to use "partitioning" to describe the most generic process of gases going into and out of particles (e.g., "gas-particle partitioning"). Partitioning can be further broken down into nonreactive- and reactive-partitioning, the former of what I would define as condensation. For the latter, I would include particle phase oligomer formation as well as salt formation as representative mechanisms. The author have their own definitions for these terms; there is a reference to Pankow's 1994 manuscript so I wonder if this is a distinction that is made therein? If so, it might be of service to others to perhaps be clearer and/or consider what I would think of as more common uses of these terms.

3/9: This sentence implies that the studies cited in the previous paragraph all point to accretion product formation in the gas phase, however this is not clear in that discussion that this was in fact the main conclusions of some of the studies such as that of Kidd et al. Please clarify.

3/24: It may help the reader appreciate this better if the actual SA:Volume ratios are stated for particles at these diameters.

4/9: I am curious as to why the authors chose only to perform these experiments under dry conditions? Surely a more atmospherically relevant RH would be closer to 40%. Can the authors comment on whether or not they feel the RH has an impact on the results of this study?

7/13: This section opens up with a statement regarding the importance of considering the gas phase products of $\beta$-pinene oxidation; however, nowhere are measurements

of gas phase products actually presented that I can discern. Most of the discussion seems to be about size-resolved particle composition. Consider modifying this opening sentence to be more representative of the subject of the section (i.e., the section title itself).

10/5: This section presents O/C and OSc vs. Carbon number as a function of mass loading, however the results are not very satisfying because wrapped up in this is the effect of increased particle size that accompanies increased mass loading. In fact, particle size may be the dominating factor leading to this "effect" of mass loading. I am not sure what to suggest here . . . clearly the title of this section is somewhat deceptive because this is really not a study of the impact of mass loading. Hopefully the authors can modify this section to actually say something about the effect of mass loading, as I think this would be truly interesting.

---

## Author Comment (AC2) · 12 Apr 2017

We thank the reviewer for insightful questions and comments that will significantly clarify and improve the manuscript. Reviewer comments (listed by page/line of the original manuscript and paraphrased by us) are given below along with our responses.

1/26: The reviewer suggests that we remove "small" from this sentence.

Author response: We will make the wording change as suggested.

2/12: The reviewer suggests our use of "condensation" and "partitioning" in the manuscript is confusing.

Author response: The reviewer is correct that absorptive "partitioning" as described in Pankow (1994) is a process that describes the movement of all molecular species between the gas and particle phases, independent of where the molecules were initially formed (gas or particle phase) and whether the specific molecules involved are nonvolatile, semivolatile or volatile. However, the distinction we find important to make in this manuscript is the difference between nonvolatile molecules that are initially formed in the gas phase and then distribute between the two phases vs. nonvolatile molecules that are initially formed in the particle phase and then distribute between the two phases. In either case, most of the material resides in the particle phase at equilibrium. The origin is important to consider when assessing particle size dependent composition, since movement of nonvolatile molecules from the gas phase to the particle phase is governed by particle surface-limited kinetics, while formation of nonvolatile molecules directly in the particle phase is governed by particle volume-limited kinetics. For this reason, we use the term "condensation" to describe specifically the process whereby a supersaturated vapor of nonvolatile molecules is formed in the gas phase, with subsequent movement to the particle phase. We will rewrite this paragraph to clarify our use of the terms and rationale for doing so, and we will make sure that wording elsewhere in the manuscript is consistent.

3/9: The reviewer would like a clarification of the origin of oligomers in the references we cite, since our text implies a gas phase origin for all of the studies cited in this paragraph.

Author response: The reviewer is correct to point out that oligomers in $\alpha$-pinene SOA (Kidd, 2014) are thought to be produced directly in the particle phase rather than the gas phase. The particle size dependence in that particular study is complicated since the particle sizes examined were much larger than those in the other studies cited in this paragraph. The main point of most of the studies cited in this paragraph is that lower volatility species are preferentially found in smaller particles as would be expected for a surface area driven process (i.e. nonvolatile molecule formation in the gas

phase with subsequent movement to the particle phase). We will reword this paragraph, being careful to maintain consistency with the nomenclature discussed in #2 above.

3/24: The reviewer suggests the addition of actual volume to surface area ratios for the particles studied.

Author response: We will modify this paragraph to include the polydisperse samples studied along with a comparison of volume to surface area ratios for the monodisperse vs. polydisperse samples. We note that the values for all samples are given in supporting information Table S1.

4/9: The reviewer asks why we didn't perform these experiments at e.g. 40% RH.

Author response: In our previous work (Tu, 2016), we found very little difference in the molecular composition of SOA from $\beta$-pinene ozonolysis that was generated with 35-70% RH vs. the conditions used in the current work. For this reason, we report this study at low RH. However, understanding how RH (as well as other experimental conditions) might quantitatively impact oligomer formation is an important topic for future study. We will provide some explanatory text in the revised manuscript.

7/13: The reviewer suggests modifying this paragraph to be more representative of the topic of this section (molecular composition of the particle phase) rather than the gas phase composition, which was not measured in this study.

Author response: We will modify this paragraph accordingly. It is important, though, to discuss what is known about gas phase products at the beginning of this section, because the gas phase products provide context for understanding the particle phase measurements we discuss later in the section.

10/5: The reviewer suggests that our emphasis on mass loading for the polydisperse aerosols is misleading, since it is really the change in the particle size distribution that is driving the molecular changes we observe.

Author response: We understand the confusion that can result from our choice of wording. We do specifically link the mass loading trends to the change in volume to surface area ratio, but this occurs near the end of the paragraph. Here and elsewhere (e.g. see #4 above), we will reword the text to make it clear that the key parameter for understanding molecular composition changes is particle volume to surface area ratio, not simply mass loading. We will also add a reference to and briefly discuss another study of ours that was recently published and addresses this same issue for polydisperse aerosol from a different chemical system (Wu and Johnston, Environmental Science and Technology, 2017, DOI: 10.1021/acs.est.7b00655).
* * *

---

## Author Response (AR1)

**Response to Reviewer Comments**

We gratefully thank the reviewers for the constructive comments and suggestions to improve our manuscript.  Reviewer comments are reproduced below and their references to page and line numbers are for the _original_ manuscript.  Author responses are given in bold and the page and line numbers in our responses refer to the _revised_ manuscript.

**Reviewer 1**

1. The referee questions the "similarity" of the size distributions for polydisperse samples a and d (Table S1 and Figure S1).

   **Author response: We recognize the confusion caused by our wording.  What we meant to say, and have modified the text accordingly on p. 4 lines 22-23, is that fine tuning of the size distribution of the control aerosol was done in order to assure that we had a sufficient aerosol mass concentration at each of the mobility sizes of interest (35, 60, 85, 110 nm) to permit chemical analysis.  Fine tuning was required because of the difficulty of generating particles at the smallest mobility diameter using our particular atomizer.**

2. The referee asks about the definitions of entries in Table S3.

   **Author response: We have modified Table S2 to include a footnote for the "Unique Molecular Formulas" column saying that these values represent the average and standard deviation from five replicate samples.  Each replicate gives slightly different results, which is why it is important to perform several replicates.  (See p. 6 lines 3-5.)  The difference among replicates arises mostly from formulas that have very low signal intensities.  We also added a new column entitled "Common Molecular Formulas" that gives the number of common formulas observed in all five replicates for a given sample type – these are the formulas that are evaluated and discussed in the results and discussion.  This will give the reader a better idea of repeatability of the experiment, and will show that most of the assigned formulas are indeed detected in all five replicates.**

3. Referee comment concerning page 7 lines 8-10 and Figures 2a and 2b.

   **Author response:  We modified the legend and caption of Figure 2 (p. 19) to clearly indicate the identities of the markers (NAMS, HRMS(+) and HRMS(-)), for both size-selected and polydisperse aerosols.  (By the way, the referee got the representations correct in their comment.)**

4. Referee comment about page 9, lines 18-22, why is it "not surprising" that there is no composition dependence for negative ions in Fig. 6b?

**Author response: On p. 19 lines 18-23, we expanded this discussion to explain our reasoning more thoroughly, while at the same time removing the "not surprising" phrase. Our group and others (e.g. Hall 2013, Tu 2016, Mutzel 2015) have noted in the past that molecular formulas obtained from negative ion spectra are generally more highly oxygenated/oxidized than those obtained from positive ion spectra. Highly oxygenated/oxidized formulas are suggestive of molecules that have very low volatilities. If essentially all of the monomers detected in negative ion spectra are nonvolatile, there will be no particle size dependence in their _relative_ ability to be incorporated into particles – all of these molecules will condense with similar probability when striking the particle surface. In contrast, the next paragraph (p. 9 line 24 to p. 10 line 2) discusses positive ion monomers, which potentially have a very wide range of volatiles – some are nonvolatile, while others are semivolatile. The relative amounts of non- vs. semi- volatile monomers will change as a function of particle size for the reasons given in this paragraph.**

5. Referee comment about Figure S2.

**Author response: The caption to Figure S2 has been modified – this spectrum is for 60 nm size-selected particles.**

6. Referee comment about Figure S3.

**Author response: Labels displaying the specific mass loading have been added.**

7. Typographical errors noted by referee:

**Author response: All will be corrected as mentioned by the reviewer: p. 1 line 29, p. 3 line 11, p. 5 line 13-14, p. 6 line 1.**

**Reviewer: 2  (comments are listed by page number / line number in the original manuscript)**

1/26:  The reviewer suggests that we remove "small" from this sentence.

> **Author response: The wording change has been made (p. 1 line 28).**

2/12: The reviewer suggests our use of "condensation" and "partitioning" in the manuscript is confusing.

> **Author response: The reviewer is correct that absorptive "partitioning" as described in Pankow (1994) is a process that describes the movement of all molecular species between the gas and particle phases, independent of where the molecules were initially formed (gas or particle phase) and whether the specific molecules involved are nonvolatile, semivolatile or volatile.  However, the distinction we find important to make in this manuscript is the difference between nonvolatile molecules that are initially formed in the gas phase and then distribute between the two phases vs. nonvolatile molecules that are initially formed in the particle phase and then distribute between the two phases.  In either case, most of the material resides in the particle phase at equilibrium.  The origin is important to consider when assessing particle size dependent composition, since movement of nonvolatile molecules from the gas phase to the particle phase is governed by particle surface-limited kinetics, while formation of nonvolatile molecules directly in the particle phase is governed by particle volume-limited kinetics.  For this reason, we use the term "condensation" to describe specifically the process whereby a supersaturated vapor of nonvolatile molecules is formed in the gas phase, with subsequent particle growth at the condensation rate.  We have rewritten this paragraph to clarify our use of the terms and rationale for doing so (p. 2, lines 12-19), and have made a similar change to the abstract (p. 1 lines 8-13).  These changes are consistent with wording elsewhere in the manuscript.**

3/9: The reviewer would like a clarification of the origin of oligomers in the references we cite, since our text implies a gas phase origin for all of the studies cited in this paragraph.

> **Author response: The reviewer is correct to point out that oligomers in α -pinene SOA (Kidd, 2014) are thought to be produced directly in the particle phase rather than the gas phase. The particle size dependence in that particular study is complicated since the particle sizes examined were much larger than those in the other studies cited in this paragraph.  The main point of most of the studies cited in this paragraph is that lower volatility species are**

**preferentially found in smaller particles as would be expected for a surface area driven process (i.e. nonvolatile molecule formation in the gas phase with subsequent movement to the particle phase). We reworded this sentence accordingly (p. 3 line 9).**

3/24: The reviewer suggests the addition of actual volume to surface area ratios for the particles studied.

**Author response: The specific line in question was modified to emphasize surface to volume ratio (p. 3 lines 29-30). We note that the ratios are given for all samples in supporting information Table S1. Also see response to reviewer comment 10/16.**

4/9: The reviewer asks why we didn't perform these experiments at e.g. 40% RH.

**Author response: In our previous work (Tu, 2016), we found very little difference in the molecular composition of SOA from $\beta$-pinene ozonolysis that was generated with 35-70% RH vs. the conditions used in the current work. For this reason, we report this study at low RH. However, understanding how RH (as well as other experimental conditions) might quantitatively impact oligomer formation is an important topic for future study. We added text to discuss this on p. 4 lines 10-12.**

7/20: The reviewer suggests modifying this paragraph to be more representative of the topic of this section (molecular composition of the particle phase) rather than the gas phase composition, which was not measured in this study.

**Author response: We modified the text on p. 7 lines 20-22 to provide a reasonable transition to this section and put the discussion of gas phase products in context.**

10/16: The reviewer suggests that our emphasis on mass loading for the polydisperse aerosols is misleading, since it is really the change in the particle size distribution that is driving the molecular changes we observe.

**Author response: We understand the confusion that can result from our choice of wording. We modified the text in the last paragraph of p. 10 (especially lines 17-20 and 26-30) to make it clear: 1) that volume to surface area ratio scales with mass loading for the aerosol generation method we used, 2) that the range of volume to surface area ratios for the**

polydisperse samples studied are similar to the monodisperse samples studied, and 3) we linked this result to a similar result from our group that was recently published (Wu and Johnston, 2017).